biotechnology/environmental science

*Bacillus thuringiensis*, cheap cultivation, vegetable waste, bioactivity assays

**Authors for correspondence:**
Xiong Guan
e-mail: guanxfafu@126.com
Zhi Chen
e-mail: chenzhi0529@163.com

This article has been edited by the Royal Society of Chemistry, including the commissioning, peer review process and editorial aspects up to the point of acceptance.

# Effect of *Bacillus thuringiensis* biomass and insecticidal activity by cultivation with vegetable wastes

Xiaohong Pan[1,2], Tengzhou Huang[1,2], Yun Fang[1,2], Wenhua Rao[1,2], Xueping Guo[1,2], Danyue Nie[1,2], Dingyang Zhang[1,2], Fang Cao[1,2], Xiong Guan[1,2] and Zhi Chen[3]

[1]State Key Laboratory of Ecological Pest Control for Fujian and Taiwan Crops, [2]Key Lab of Biopesticide and Chemical Biology, Ministry of Education, College of Plant Protection, and [3]College of Resources and Environmental Sciences, Fujian Agriculture and Forestry University, Fuzhou, Fujian 350002, People's Republic of China

XG, 0000-0003-3505-7464

*Bacillus thuringiensis* (Bt) has been regarded as a biopesticide with high efficiency and safety, while it still cannot be popularized and mass-produced because of its high production costs. In the present study, we aimed to develop a cost-effective biopesticide via the secondary use of discharged vegetable wastes as the raw fermentation medium, and the insecticidal activity of Bt strain prepared by this cheap cultivation approach was evaluated. The suitable carbon source, nitrogen source additives and optimal metal ions were screened by the single-factor test, and the optimal combination of additives was determined by orthogonal test and ANOVA analysis. We found that soluble starch ($6\,\text{g}\,\text{l}^{-1}$), soya bean meal ($6\,\text{g}\,\text{l}^{-1}$), $Al^{3+}$ ($0.4\,\text{g}\,\text{l}^{-1}$) and $Fe^{2+}$ ($0.4\,\text{g}\,\text{l}^{-1}$) were the optimal exogenous additives, and the optimal fermentation conditions were as follows: pH 7.0, temperature of 35°C and aeration of 80 ml/ 250 ml. Meanwhile, the bioactivity test results showed that the Bt strain prepared by cheap cultivation still exhibited a good insecticidal effect on *Helicoverpa armigera* compared with the standard LB medium. Collectively, our findings provided a new strategy for vegetable waste utilization with less environmental impact and reduced production cost.

# 1 Introduction

As an effective and safe biological insecticide, *Bacillus thuringiensis* (Bt) is effective against lepidopteran, coleopteran, dipteran insect

pests and nematodes [1,2]. The industrial medium for bacterial cultivation is relatively expensive, and its main components include peptone, yeast extract, sodium chloride, $K_2HPO_4$ and other ingredients [3]. It has been estimated that the price of fermentation medium accounts for 35–59% of the production cost [4]. Consequently, the application of Bt insecticide is rare (only from 2% [5] to 6% [6] of the insecticidal market) because of the high cost in the fermentation process. Therefore, how to reduce the fermentation cost of Bt is of great importance from both economical and practical points of view.

Generally speaking, the agroindustry wastes are continuously produced at a global level, and it can reach a level of billion tonnes every year [7]. Currently, various agro-industrial wastes, such as rice straw, poultry litter, wheat bran and spent mushroom substrate (SMS), have been successfully used as a costless fermentation medium to reduce the cost of Bt production [4,8]. Among these agroindustry wastes, the vegetable wastes contain high contents of water and rich organic compounds, such as carbohydrates, lipids, starch, cellulose and organic acids [9,10], and this nutritional ingredient would ensure the normal growth, sporulation and crystal formation of Bt. It was reported that the global production of fruit and vegetables could reach more than 300 million tons per year, and the improper disposal methods for vegetable wastes could cause negative environmental issues, such as ground pollution and greenhouse gas emissions because of their high organic matter contents [11,12]. Moreover, the vegetable wastes are mainly produced in planting fields and processing and trading places, which can easily be collected separately. However, to the best of our knowledge, there are no reports on the application of vegetable wastes for Bt fermentation. The complete use of discharged vegetable wastes not only reduces the cost of Bt fermentation medium but also hinders the possible pollution, by which double economic and environmental benefits can be achieved.

In the present study, we aimed to exploit the discharged vegetable wastes as the raw fermentation medium of Bt, the growth conditions (such as pH value, additional nutrients, temperature and aeration) were optimized for bacterial growth, and the insecticidal activity of Bt strain prepared by cheap cultivation was evaluated. Collectively, our findings provided a new strategy for vegetable waste utilization with less environmental impact and reduced production cost.

# 2. Material and methods

## 2.1. The bacterial growth conditions and bacterial biomass analysis

*Bacillus thuringiensis* subsp. *kurstaki* 8010 (serotype 3a3b) used in this study was isolated from dead larva of *Papilio polytes* L. [13], which is a commercial strain production of Cry1Ab protein (130 kDa). Bacteria were inoculated at a ratio of 1% into 100 ml medium, followed by incubation at 37°C with agitation at 160 r.p.m. The common laboratory liquid medium (LB medium, set as the positive group) consisted of peptone 10 g l$^{-1}$, yeast extract 5 g l$^{-1}$, sodium chloride 10 g l$^{-1}$ and the pH is adjusted to 7.0. After 24 h of incubation, the growth of Bt was monitored by UV/VIS spectrometer at a wavelength of 600 nm.

## 2.2. Screening and preparation of vegetable wastes

The vegetable wastes were obtained from a farmer's market in Fuzhou City (Fujian Province, China). Vegetable wastes were mainly composed of tomato, Chinese cabbage, green beans, cabbage and spinach. The preparation process of vegetable wastes supernatant was contained washing, pressing, filtration, centrifugation and sterilization. Briefly, the vegetable wastes were washing cleanly by distilled water, then the viscous juice was obtained by pressing and filtration of the vegetable wastes. Subsequently, the supernatant was obtained by centrifugation at 13 000g for 10 min, sterilized at 121°C for 20 min, and then stored at −20°C before further analysis.

## 2.3. The optimal conditions for bacterial growth

### 2.3.1. The suitable carbon source, nitrogen source additives and optimal metal ions by single-factor test

Glucose, sucrose, soluble starch and mannitol were selected as the experimental carbon sources, and the concentration was set at 6 g l$^{-1}$. Tryptone, corn starch, urea and soya bean meal were selected as the experimental nitrogen sources, and the concentration was also set at 6 g l$^{-1}$. Moreover, $MgSO_4$, $FeSO_4 \cdot 7H_2O$, $Al_2(SO_4)_3$ and NaCl were designated as the additional mineral elements, and the

concentration was $0.4 \, g \, l^{-1}$. The suitable carbon source, nitrogen source additives and optimal metal ions were determined by a single-factor test. Moreover, the conventionally used LB medium for Bt cultivation was added as a positive control and compared to treatment.

### 2.3.2. The optimal concentration of additives by orthogonal test

According to the result of the single-factor test, the soya bean cake powder, soluble starch, as well as $Al^{3+}$, and $Fe^{2+}$ were selected as the additional carbon sources, nitrogen source and metal ions, respectively. Subsequently, the orthogonal experiment was carried out by using SPSS17 software with different concentration gradients (2, 4, 6% for carbon source; 2, 4, 6% for nitrogen source and 0.4, 0.8, 1.2% for metal ions). A general linear model was established by using L9 (four factors and three levels) orthonormal table, the significance difference of each factor was analysed, then the optimal concentration was determined through variance analysis and Duncan's multiple comparison, and the OD value was intuitively analysed based on the mean of each level calculated from L9 orthogonal results.

### 2.3.3. The final optimized fermentation conditions for bacterial growth

The initial pH of the vegetable wastes medium ranged from 5 to 9. After 1% inoculation, the bacteria were incubated in a shaker at a constant temperature of 37°C for 72 h with agitation at 150 r.p.m., and then the bacterial biomass was evaluated to determine the optimal pH for Bt growth. Similarly, the incubation temperature of constant temperature shaker was adjusted to 15°C, 20°C, 25°C, 30°C and 35°C for determination of the optimal fermentation temperature. Meanwhile, the optimal ventilation rate was also measured by adjusting the liquid volume of 40, 60, 80, 100, 140 and 160 ml in a 250 ml conical flask. All the treatments were carried out in triplicate.

## 2.4. The insecticidal activity of Bt in the optimized cheap medium

Bioassays were conducted according to a previously described method by Rao *et al.* with minor modifications [14]. Briefly, the prepared artificial diet was added into 24-well plates, then 100 µl of bacterial suspension was evenly transferred onto the surface of the diet, and it was allowed to dry in the clean bench before the addition of the larvae. Subsequently, one first-instar *Helicoverpa armigera* was added to each well ($n = 24$/treatment), encapsulated and incubated in an incubator ($T = 37 \pm 1$°C, RH $= 60 \pm 5\%$, L : D $= 16 : 8$). Finally, the larval mortality was calculated every 24 h. The common LB medium for Bt cultivation served as the control. All the treatments were carried out in triplicate. The significant difference was analysed with SPSS17 by one-way ANOVA analysis, and data represent mean $\pm$ s.d. $p < 0.05$ was considered as statistically significant.

# 3. Results

## 3.1. Effects of different vegetable wastes on bacterial growth

Figure 1*a* shows that cabbage and spinach had similar effects on Bt growth, and tomato could not promote the effective growth of Bt since the tomato medium led to the production of more sediment during the experimental process. Moreover, green beans and Chinese cabbage could better promote the growth of Bt strains compared with the tomato, spinach and cabbage, and the green beans showed the highest OD value. However, the green beans were more expensive compared with Chinese cabbage, and green beans came from relatively few sources compared with Chinese cabbage. Therefore, we selected the mixture of green bean and Chinese cabbage as the growth medium for Bt cultivation in subsequent experiments due to the consideration of cost and source.

Additionally, the Bt strain was grown well in the nutritious LB medium, but Bt strain also maintained a relatively high $OD_{600}$ value in the mixture of green bean and Chinese cabbage (figure 1*b*), and the green beans and Chinese cabbage at a ratio of 2 : 1 were chosen as the optimum medium for the growth of Bt. Previous studies have indicated that the vegetable wastes contain high contents of water and rich organic compounds [9,10], implying that vegetable wastes can provide the possible carbon source and nitrogen source for Bt cultivation.

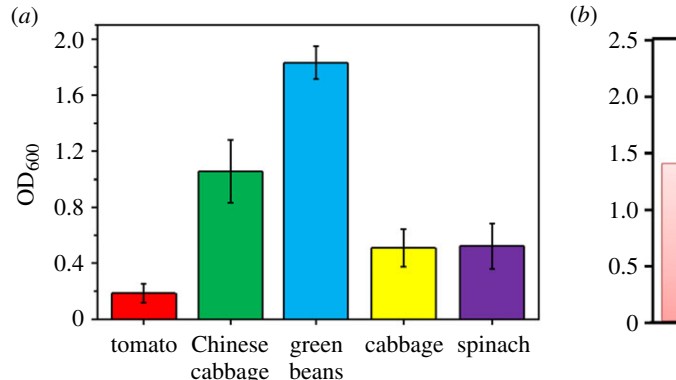
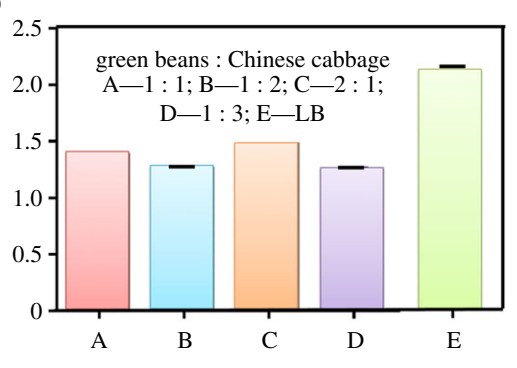

**Figure 1.** The effects of different vegetable wastes on bacterial growth. (*a*) Different vegetable wastes and (*b*) the different ratio of green beans and Chinese cabbage.

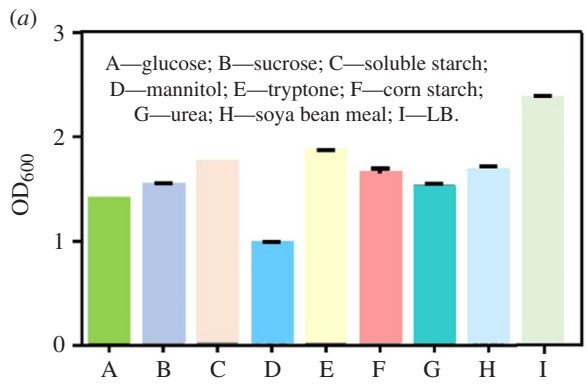
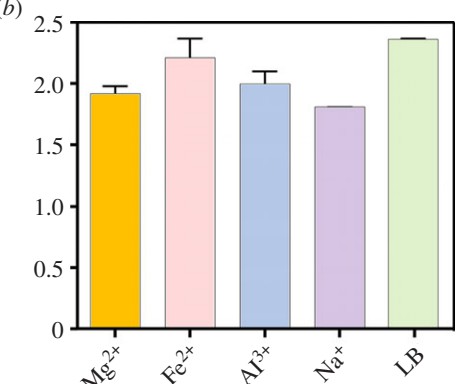

**Figure 2.** (*a*) The growth of Bt in vegetable waste medium with different carbon sources (glucose, sucrose, soluble starch and mannitol) and nitrogen sources (tryptone, corn starch, urea and soya bean meal). (*b*) Addition of different metal ions in vegetable waste medium with soluble starch and soya bean meal. The LB medium is set as positive control.

## 3.2. Effects of carbon source, nitrogen source and metal ion additives on bacterial growth

To promote the growth of Bt bacteria in the vegetable waste medium, four carbon sources (glucose, sucrose, soluble starch and mannitol) and four nitrogen sources (tryptone, corn starch, urea and soya bean meal), with concentrations of $6 \, g \, l^{-1}$, were added to the pre-treated vegetable waste culture medium, respectively. The results also showed that the Bt strain was grown well in the nutritious LB medium. However, mannitol could not significantly promote the growth of the bacterial strain as the carbon source additive, and the other three additives had relatively small differences in the growth of Bt (figure 2*a*). Meanwhile, the soluble starch was more effective in accelerating bacterial growth. Therefore, soluble starch was selected as the carbon source additive. As for nitrogen source additives (figure 2*a*), urea had a small promotive effect on the growth of Bt, which might be attributed to that it was an inorganic nitrogen source. Both tryptone and soya bean meal played a positive role in promoting Bt growth, and there was no difference between these two. However, we chose soya bean meal as the nitrogen source additive, since the cost of tryptone was higher compared with soya bean meal. Besides, compared with the positive control group using LB medium for fermentation, we found that adding carbon source or nitrogen source alone could not achieve a promotive effect on Bt growth. Therefore, it is necessary to consider the combination of carbon source and nitrogen source additives to achieve a better effect on the growth of bacterial strains.

Inorganic ions play an important role in the growth of Bt, such as cell composition and energy transfer. Based on the above-mentioned results, four commonly used metal ions ($Mg^{2+}$, $Fe^{2+}$, $Al^{3+}$ and $Na^+$) were, respectively, added into the pretreatment medium of vegetable waste, the addition amount was $1.2 \, g \, l^{-1}$ and the growth effects were compared with the LB medium positive group (figure 2*b*). It was found that the pretreatment medium supplemented with $Fe^{2+}$ had the best growth effect, the fermentation level was close to that of LB medium, and the promotive effect was better than other metal ions, followed by $Al^{3+}$. Therefore, $Fe^{2+}$ and $Al^{3+}$ were selected as metal ion additives.

**Table 1.** Experimental design of the four controlling factors with three levels.

| levels | A soluble starch (%) | B soya bean meal (%) | C Fe$^{2+}$ (%) | D Al$^{3+}$ (%) |
|---|---|---|---|---|
| 1 | 2 | 2 | 0.4 | 0.4 |
| 2 | 4 | 4 | 0.8 | 0.8 |
| 3 | 6 | 6 | 1.2 | 1.2 |

**Table 2.** L9 orthogonal array for mixed design.

| run | level | | | | OD value |
|---|---|---|---|---|---|
| | A | B | C | D | |
| 1 | 1 | 1 | 1 | 1 | 1.997 |
| 2 | 1 | 2 | 2 | 2 | 2.069 |
| 3 | 1 | 3 | 3 | 3 | 2.291 |
| 4 | 2 | 1 | 2 | 3 | 2.050 |
| 5 | 2 | 2 | 3 | 1 | 2.073 |
| 6 | 2 | 3 | 1 | 2 | 2.282 |
| 7 | 3 | 1 | 3 | 2 | 2.163 |
| 8 | 3 | 2 | 1 | 3 | 2.334 |
| 9 | 3 | 3 | 2 | 1 | 2.553 |

## 3.3. The optimal combination of carbon source, nitrogen source and metal ion additives by orthogonal test

To improve the fermentation production of Bt in the vegetable wastes, it is necessary to optimize the medium based on exogenous additives by orthogonal test. Carbon source (soluble starch), nitrogen source (soya bean meal) and metal ions (Fe$^{2+}$ and Al$^{3+}$) were added into the pre-treated vegetable waste medium in different proportions by four-factor and three-level test, and the optimal combinations were selected. Four experimental factors were A (soluble starch), B (soya bean meal), C (Fe$^{2+}$) and D (Al$^{3+}$); table 1 lists the levels of these four factors. Table 2 describes the L9 orthogonal array for the mixed design, then batch tests were conducted to determine the significance of each factor, and the OD value was adopted to evaluate the influence among four experimental factors. Figure 3 shows the OD value (mean of each level calculated based on table 2) of the four factors, and we found that the optimal condition was A$_3$B$_3$C$_2$D$_3$, indicating that the optimal composition of the medium was 6 g l$^{-1}$ soluble starch, 6 g l$^{-1}$ soya bean meal, 0.4 g l$^{-1}$ Fe$^{2+}$ and 0.4 g l$^{-1}$ Al$^{3+}$. Meanwhile, an analysis of variance (ANOVA) was performed, and the results are described in table 3. The results indicated that soluble starch and soya bean meal had obvious significance ($p < 0.05$) by variance analysis, implying that soluble starch and soya bean meal had a significant effect on bacterial growth. Moreover, the fermentation production of Bt in the optimal medium was equivalent to LB medium, and it could significantly promote bacterial growth compared with the other three randomly selected media.

## 3.4. Effects of pH, temperature and aeration on bacterial growth

The culture pH is an important parameter because it can affect the charge of the cell surface [15], and the growth and metabolism of microorganisms are directly related to pH value. As a result, the effect of pH value on bacterial growth was evaluated. Figure 4a shows that the bacterial concentration was the lowest in the medium with pH 5.0 and 9.0, implying that both the acidic and alkaline conditions would inhibit the bacterial growth, and the optimal initial pH range was from 7.0 to 7.5 for bacterial growth.

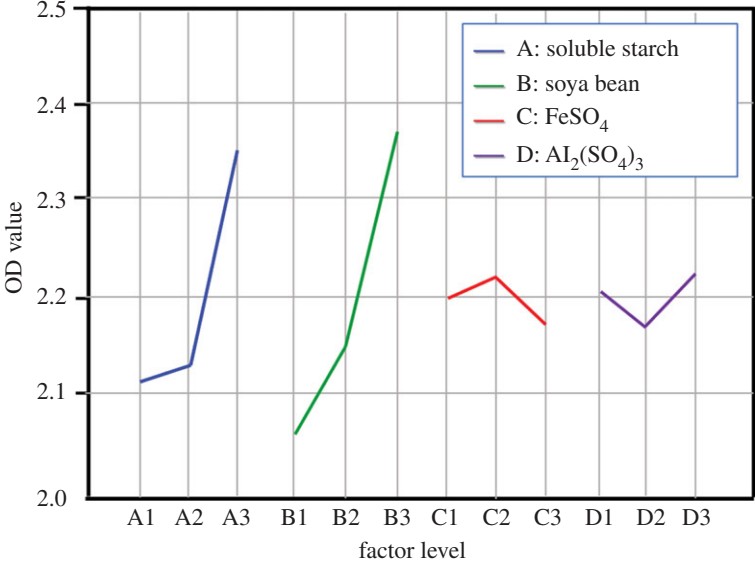

**Figure 3.** The intuitive analysis of the orthogonal experiment, the OD value of four factors (A, soluble starch; B, soya bean meal C, $Fe^{2+}$; D, $Al^{3+}$) at three level is calculated based on table 2.

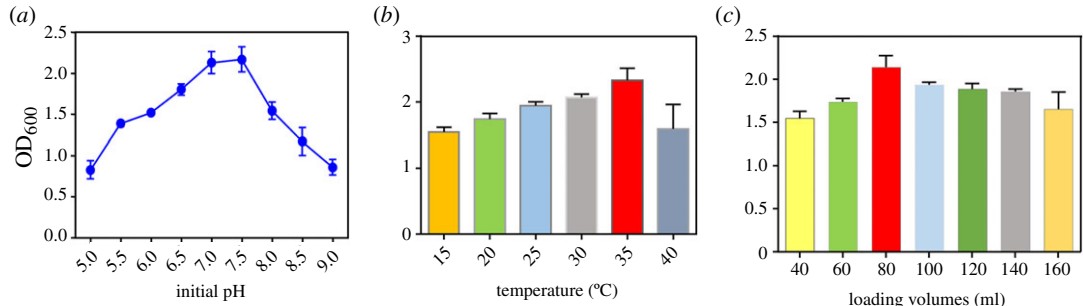

**Figure 4.** Effects of different initial pH of vegetable waste medium (*a*), fermentation temperature (*b*) and aeration condition (*c*) on the growth of Bt.

**Table 3.** ANOVA results of orthogonal experimental design on culture medium's additives by validation variance. Factors: A, soluble starch; B, soya bean meal; C, $Fe^{2+}$; D, $Al^{3+}$.

|  | sum of squares | degrees of freedom | mean square | computed $F$-value | significance ($p$-value) |
|---|---|---|---|---|---|
| intercept | 87.212 | 1 | 87.212 | 76 505.855 | 0.000 |
| A | 0.200 | 2 | 0.100 | 87.585 | *0.000* |
| B | 0.296 | 2 | 0.148 | 129.974 | *0.000* |
| C | 0.007 | 2 | 0.004 | 3.110 | 0.094 |
| D | 0.009 | 2 | 0.005 | 3.948 | 0.059 |
| error | 0.01 | 9 | 0.001 |  |  |

Meanwhile, the fermentation temperature would be a critical factor for the metabolic rate and bioactivity of Bt. Therefore, we also monitored the fermentation temperature (figure 4*b*). We found that the growth rate of bacteria was gradually increased with the increase in fermentation temperature, the $OD_{600}$ value of Bt was the highest at 35°C, and the higher temperature was not good for the growth of bacteria. Previous studies have also indicated that the Bt bacteria experience premature ageing at the high temperature, and the virulence of Bt is decreased. On the contrary, the

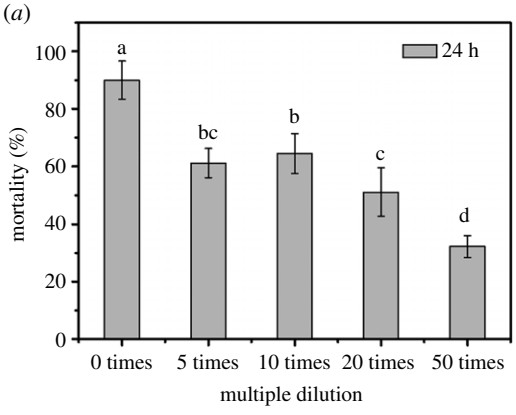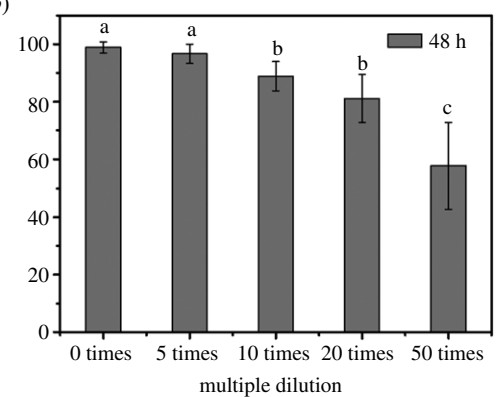

**Figure 5.** The insecticidal activity on first-instar *H. armigera* at 24 h (*a*) and 48 h (*b*). The multiple of fermentation liquid is from 0 to 50 times. Different letters within figure denote a significant difference ($p < 0.05$).

low temperature leads to slow growth of bacteria, prolongs the fermentation period and decreases the bacterial virulence [16]. Therefore, suitable temperature is essential for Bt growth.

Oxygen plays an important role in the fermentation process, and the oxygen supply is closely related to the yields [17]. Since Bt is an aerobic bacterium, the aeration capacity of the medium can ensure its normal growth and metabolism. The results indicated that the bacteria grew well when the loading volume was 80 ml/250 ml (figure 4*c*). It has been reported that the sporulation of Bt is highly related to $O_2$ supply, and the Bt fails to survive or sporulate at low aeration levels, especially in the presence of high concentrations of sugar [18].

## 3.5. The insecticidal activity of Bt in the optimized vegetable waste medium

In the present study, we measured the biological activity of Bt in the optimized medium. Figure 5 shows that Bt 8010 strain fermented in the vegetable waste medium had a good insecticidal effect, the mortality could reach 96.7% by five times of dilution after 48 h of feeding, and the mortality was equally effective compared with the laboratory standard medium. Moreover, similar to the LB control group, the high mortality at vegetable waste medium in a short time might be attributed to the multiple dilution, and it was gradually decreased when the dilution ratio was increased, while the diluted bacterial solution (50 times) still exhibited good insecticidal effect (57.8%) after 48 h of feeding. It was noted that the $LC_{50}$ of standard LB medium (positive group) and our optimized medium was 16.783 and 17.420 µl ml$^{-1}$, respectively. The results indicated that Bt could effectively use the vegetable waste medium and maintain its good insecticidal activity.

# 4. Discussion

The sources of vegetable wastes are widespread and mainly generated in wholesale markets, supermarkets and agricultural activities [19]. The vegetable wastes constitute an important potential source for valuable natural products and chemicals, such as enzymes, exopolysaccharides, bioplastics and biofuels [20]. However, the improper disposal of vegetable wastes may lead to negative environmental issues. Hence, the utilization of vegetable waste would be socially useful and environmentally beneficial. Wu *et al*. [8] have used the SMS extract as a potential carbon source for Bt culture, while the process of pretreatment and enzymatic hydrolysis is relatively complicated. Meanwhile, few reports have shown that the vegetable wastes can be used as a cheap medium for bacterial cultivation.

In the present study, Bt could effectively use the vegetable wastes for bacterial growth with a simple pretreatment process, the additional nutrients (6 g l$^{-1}$ soluble starch, 6 g l$^{-1}$ soya bean meal, 0.4 g l$^{-1}$ Al$^{3+}$, and 0.4 g l$^{-1}$ Fe$^{2+}$) would accelerate the growth of bacteria, the bacteria could be well grown at neutral pH, the optimal fermentation temperature was 35°C and the optimal aeration condition was 80 ml/250 ml. The biological activity experiment indicated that Bt could effectively use the vegetable waste medium and maintain its good insecticidal activity. Therefore, the vegetable wastes could be used as a cost-effective medium for the cultivation of Bt, which would be greatly beneficial not only for biopesticide production but also for reducing the environmental pollution of agricultural waste.

Data accessibility. All original data are available from the Dryad Digital Repository: https://doi.org/10.5061/dryad.98sf7m0gw [21].

Authors' contributions. X.P. designed the study, analysed the data and wrote the text. T.H., Y.F., W.R., X.G. and D.N. performed the optimized conditions for bacterial growth studies. D.Z. and F.C. undertook the insecticidal activity of Bt in the optimized cheap medium. Z.C. and X.G. participated in the manuscript preparation. All authors contributed to its final form and gave final approval for its publication.

Competing interests. We declare we have no competing interests.

Funding. X.G. was supported by the National Key R&D Program of China (grant no. 2017YFE0121700), Special Fund for Scientific and Technological Innovation of Fujian Agriculture and Forestry University (grant nos. KHF190013 and KF2015064-065). X.P. was supported by the Natural Science Foundation of Fujian Province, China (grant no. 2020J01522), Science Fund for Distinguished Young Scholars of Fujian Agriculture and Forestry University (grant no. xjq201719) and the Special Fund for Scientific and Technological Innovation of Fujian Agriculture and Forestry University (grant no. CXZX2019005S).

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
