## [Peer Review File · Royal Society Open Science]

Review History

RSOS-201564.R0 (Original submission)

Review form: Reviewer 1

Is the manuscript scientifically sound in its present form?

Yes

Are the interpretations and conclusions justified by the results?

Yes

Is the language acceptable?

Yes

Do you have any ethical concerns with this paper?

No

Have you any concerns about statistical analyses in this paper?

No

Recommendation?

Accept with minor revision (please list in comments)

Comments to the Author(s)

This manuscript describes the investigation of the Effect of *Bacillus thuringiensis* biomass and insecticidal activity by cultivation with vegetable wastes. The topics presented in this manuscript are scientifically meaningful and interesting. The data is very comprehensive, but some fundamental issues could be clarified prior to evaluating the values of the current works.

My concerns about this manuscript are as follows:

1. Why the author selected vegetable wastes as cheap culture, rather than other agroindustry wastes? The authors should comment on this.
2. What is the contribution of vegetable waste to *Bacillus thuringiensis* (Bt) culture?
3. All the results of the culture experiments presented should be explained in comparison with the positive control or normal condition.
4. The significance analysis would be added in Figure 5.

Review form: Reviewer 2 (Ricardo Polancyck)

Is the manuscript scientifically sound in its present form?

No

Are the interpretations and conclusions justified by the results?

No

Is the language acceptable?

Yes

Do you have any ethical concerns with this paper?

No

Have you any concerns about statistical analyses in this paper?

No

Recommendation?

Major revision is needed (please make suggestions in comments)

Comments to the Author(s)

The authors should added simple PCR analyses from the cultivated Bt with the goal to described which toxins (Cry or Vip) are account for the *Helicoverpa armigera* mortality. Mortality data from the control should be added, as well. It's not common to Bt a such high mortality (85%) in the first day evaluation. One possibility is the presence of undesirable toxins in the product, such as exotoxins - this must be evaluated.

Otherwise the authors should provided a detailed decription (line 65) about the material used fot Bt cultivation and how these material was prepared because it's possible the presence of Bt and other entomopathogens in the surface of this material. Maybe just one washing is not enough to clean the material surface.

Line 35 the authors must be update the only 2% of the insecticidal market. Recent data published by Rao & Jurat-Fuentes (2019) indicate that this value is about 6%.

Decision letter (RSOS-201564.R0)

Dear Professor Guan:

Title: Effect of *Bacillus thuringiensis* biomass and insecticidal activity by cultivation with vegetable wastes
Manuscript ID: RSOS-201564

The editor assigned to your manuscript has now received comments from reviewers. We would like you to revise your paper in accordance with the referee and Subject Editor suggestions which can be found below (not including confidential reports to the Editor). Please note this decision does not guarantee eventual acceptance.

Please submit your revised paper before 11-Dec-2020. Please note that the revision deadline will expire at 00.00am on this date. If we do not hear from you within this time then it will be assumed that the paper has been withdrawn. In exceptional circumstances, extensions may be possible if agreed with the Editorial Office in advance. We do not allow multiple rounds of revision so we urge you to make every effort to fully address all of the comments at this stage. If deemed necessary by the Editors, your manuscript will be sent back to one or more of the original reviewers for assessment. If the original reviewers are not available we may invite new reviewers.

Royal Society of Chemistry
Thomas Graham House
Science Park, Milton Road
Cambridge, CB4 0WF

Royal Society Open Science - Chemistry Editorial Office

On behalf of the Subject Editor Professor Anthony Stace and the Associate Editor Dr Nadia Martinez Villegas.

RSC Associate Editor:

Comments to the Author:

The research presented in this draft is original and of interest to RSOS audience, however additional analyses are needed to discuss culture experiment results in comparison with the positive control and explain the high mortality. Additionally, the Material and Methods section and the quality of the presentation of the figures should be improved.

RSC Subject Editor:

Comments to the Author:

(There are no comments.)

Reviewers' Comments to Author:

Reviewer: 1

Comments to the Author(s)

This manuscript describes the investigation of the Effect of *Bacillus thuringiensis* biomass and insecticidal activity by cultivation with vegetable wastes. The topics presented in this manuscript are scientifically meaningful and interesting. The data is very comprehensive, but some fundamental issues could be clarified prior to evaluating the values of the current works. My concerns about this manuscript are as follows:

1. Why the author selected vegetable wastes as cheap culture, rather than other agroindustry wastes? The authors should comment on this.
2. What is the contribution of vegetable waste to *Bacillus thuringiensis* (Bt) culture?
3. All the results of the culture experiments presented should be explained in comparison with the positive control or normal condition.
4. The significance analysis would be added in Figure 5.

Reviewer: 2

Comments to the Author(s)

The authors should added simple PCR analyses from the cultivated Bt with the goal to described which toxins (Cry or Vip) are account for the *Helicoverpa armigera* mortality. Mortality data from the control should be added, as well. It's not common to Bt a such high mortality (85%) in the first day evaluation. One possibility is the presence of undesirable toxins in the product, such as exotoxins - this must be evaluated.

Otherwise the authors should provided a detailed decription (line 65) about the material used fot Bt cultivation and how these material was prepared because it's possible the presence of Bt and other entomopathogens in the surface of this material. Maybe just one washing is not enough to clean the material surface.

Line 35 the authors must be update the only 2% of the insecticidal market. Recent data published by Rao & Jurat-Fuentes (2019) indicate that this value is about 6%.

Author's Response to Decision Letter for (RSOS-201564.R0)

See Appendix A.

RSOS-201564.R1 (Revision)

Review form: Reviewer 1

Is the manuscript scientifically sound in its present form?

Yes

Are the interpretations and conclusions justified by the results?

Yes

Is the language acceptable?

Yes

Do you have any ethical concerns with this paper?

No

Have you any concerns about statistical analyses in this paper?

No

Recommendation?

Accept as is

Comments to the Author(s)

All my concerns have been addressed. I recommend an acceptance of this paper.

Decision letter (RSOS-201564.R1)

Dear Professor Guan:

Title: Effect of *Bacillus thuringiensis* biomass and insecticidal activity by cultivation with vegetable wastes

Manuscript ID: RSOS-201564.R1

It is a pleasure to accept your manuscript in its current form for publication in Royal Society Open Science. The chemistry content of Royal Society Open Science is published in collaboration with the Royal Society of Chemistry. I apologise it has taken much longer than usual to be able to send you this decision.

On behalf of the Subject Editor Professor Anthony Stace and the Associate Editor Dr Nadia Martinez Villegas.

RSC Associate Editor:
Comments to the Author:
Thank you very much for the revised version of the manuscript. Changes were made up to the satisfaction of the reviewers.

RSC Subject Editor:
Comments to the Author:
(There are no comments.)

Reviewer(s)' Comments to Author:
Reviewer: 1

Comments to the Author(s)
All my concerns have been addressed. I recommend an acceptance of this paper.

Appendix A

Point-by-point response to editor and reviewers

RSC Associate Editor's comments:

The research presented in this draft is original and of interest to RSOS audience, however additional analyses are needed to discuss culture experiment results in comparison with the positive control and explain the high mortality. Additionally, the Material and Methods section and the quality of the presentation of the figures should be improved.

Answer: We appreciate the valuable suggestions and time spend of you and reviewers. The manuscript has been carefully revised according to the referee reports, and the point-by-point response to referees has been attached.

Moreover, according to your suggestion, we have added the discussion on the culture experiment comparison with the positive control and the reason on the high mortality, the Material and Methods section, the quality of the presentation of the figures and captions was improved in the revised manuscript.

We hope these revisions will make our manuscript more acceptable for publication. Thanks for your consideration.

=====

Reviewers' Comments

Reviewer: 1

This manuscript describes the investigation of the Effect of *Bacillus thuringiensis* biomass and insecticidal activity by cultivation with vegetable wastes. The topics presented in this manuscript are scientifically meaningful and interesting. The data is very comprehensive, but some fundamental issues could be clarified prior to evaluating the values of the current works.

My concerns about this manuscript are as follows:

1. Why the author selected vegetable wastes as cheap culture, rather than other agroindustry wastes? The authors should comment on this.

Answer: Thanks for your valuable suggestions. It was reported that the global production of fruit and vegetables could reach more than 300 million tons per year, and the improper disposal methods for vegetable wastes could cause negative environmental issues, such as ground pollution and greenhouse gas emissions because of their high organic matter contents. Moreover, the vegetable wastes are mainly produced in planting fields and processing and trading places, which are easily to be collected separately. Therefore, we selected vegetable wastes as cheap culture for Bt fermentation. As suggested, we have added the related explanation in the revised manuscript in Introduction Section (**Please see Page 3, Line 44-49**).

2. What is the contribution of vegetable waste to Bacillus thuringiensis (Bt) culture?

Answer: Thanks. The vegetable wastes containing high water content and rich organic compounds, such as carbohydrates, lipids, starch, cellulose and organic acids, and this nutritional ingredient would ensure the normal growth, sporulation and crystal formation of Bt (**Page 3, Line 41-44**). Moreover, previous studies also

indicated that the vegetable wastes contains high content of water content and rich organic compounds, which implies that vegetable wastes could be provided the possible carbon source and nitrogen source for Bt cultivation (Page 7, Line 128-130).

3. All the results of the culture experiments presented should be explained in comparison with the positive control or normal condition.

Answer: Thanks for your comments. The conventional cultivation of Bt was selected as the positive control (using LB medium for Bt fermentation), and we have added the discussion on the comparison with positive control and treatment experiments in the revised manuscript as suggested. More details could be seen in in the revised manuscript (Page 4-11, Line 64-205, marked with yellow color).

4. The significance analysis would be added in Figure 5.

Answer: Thanks for pointing out this. The significance analysis was added in the Figure 5.

=====

Reviewer: 2

1. The authors should added simple PCR analyses from the cultivated Bt with the goal to described which toxins (Cry or Vip) are account for the Helicoverpa armigera mortality. Mortality data from the control should be added, as well. It's not common to Bt a such high mortality (85%) in the first day evaluation. One possibility is the presence of undesirable toxins in the product, such as exotoxins - this must be evaluated.

Answer: Sorry for the confusion. *Bacillus thuringiensis* subsp. *kurstaki* 8010 (serotype 3a3b) used in this study was isolated from dead larva of *Papilio polytes* L., which is a commercial strain production of Cry1Ab protein (130 kDa), thus the Cry toxins would be account for the *H. armigera* mortality. Moreover, similarly to the LB control group, the high mortality at vegetable waste medium might attribute to the multiple dilution, the mortality rate of Bt 8010 strain fermentation liquid was gradually decreased when the dilution ratio was increased.

According to your comment, we have added the related Bt strain information and added the possible reason on the high mortality in the revised manuscript (please see Page 4, Line 61-63 and Page 11, Line 201-203).

2. Otherwise the authors should provided a detailed decription (line 65) about the material used fot Bt cultivation and how these material was prepared because it's possible the presence of Bt and other entomopathogens in the surface of this material. Maybe just one washing is not enough to clean the material surface.

Answer: Thanks for your comments. The preparation process of vegetable wastes supernatant was not only contained washing, pressing and filtration, but also obtained by centrifugation at 13000g for 10 min and sterilized at 121 °C for 20 min. Therefore, other entomopathogens would not exist in the surface of this material. And we have added more details on the preparation process, please see Page 5, Line 71-75.

3. Line 35 the authors must be update the only 2% of the insecticidal market. Recent data published by Rao & Jurat-Fuentes (2019) indicate that this value is about 6%.

Answer: Thanks for pointing out this. We have updated the data in the revised manuscript and added the related reference. Please see in **Page 3, Line 35** and **Reference 6**.